# Consistent Theories of Free Dirac Particle without Singular Predictions

Giuseppe Nisticò [1,2] 

1 Dipartimento di Matematica e Informatica, Università della Calabria, Via P. Bucci, 30B, I-87036 Rende, Italy; giuseppe.nistico@unical.it; Tel.: +39-(0)-984-496413
2 INFN, Gr. Collegato di Cosenza, I-87036 Rende, Italy

**Abstract:** Dirac's theory is not a unique theory consistent with the physical principles specific of a free spin-one-half particle. In fact, we derive classes of theories of an elementary free particle from the principle of Poincaré's invariance and from the principle of the covariance of the position. The theory of Dirac is just one of these theories, characterized by singular predictions, namely, the zitterbewegung. Yet, the class here derived contains families of consistent theories without singular predictions. For the time being, the experimental verifiability of these alternative theories is restricted to the predictions of free-particle theories for ideal experiments.

**Keywords:** Dirac particle; Dirac's theory; zitterbewegung; consistent theories without singular features



## 1. Introduction

The quantum theory of spin-one-half particles formulated by Dirac [1,2] suffered several problems, effectively reviewed in [3]. The difficulty encountered in trying to solve these problems has its roots in the features of the method Dirac used to build up his theory, the method of *canonical quantization*. This method prescribes to start from the *classical theory* of the system under investigation, e.g., for a particle, from the Hamiltonian formulation of its theory with conjugate dynamical variables $q_j$ and $p_j$ and Hamiltonian $h(\mathbf{q}, \mathbf{p}, t)$; then, it is dictated to replace the variables $q_j$ and $p_j$ with operators $Q_j$ and $P_j$ of a Hilbert space and the Poisson brackets $\{f(\mathbf{q}, \mathbf{p}), g(\mathbf{q}, \mathbf{p})\}$ with the operator commutators $i[f(\mathbf{Q}, \mathbf{P}), g(\mathbf{Q}, \mathbf{P})]$ in the equations of the classical theory, obtaining as a result the equations of the quantum theory of the system, such as, for instance, the equation $\frac{d\mathbf{Q}}{dt} \equiv \dot{\mathbf{Q}} = i[h(\mathbf{Q}, \mathbf{P}, t), \mathbf{Q}]$. Hence, canonical quantization is a procedure to build up *models* of specific quantum theories, rather than a method that deductively derives the theory from sound basic principles.

The lack of a deductive path makes it ultimately very hard to identify the causes of the difficulties encountered by Dirac's theory. One extensively debated problem is the fact that Dirac's theory implies a very singular prediction: the time derivative $\dot{Q}_j$ of each component $Q_j$ of the position operator has its absolute value equal to the speed of light. Moreover, detailed studies of the solutions of the free Dirac equation predict violent oscillations of the particle, a phenomenon called the *zitterbewegung* [4,5], which is a very singular behaviour for a free particle, though it does not affect the consistency of the theory. Many investigations were carried out to clarify the features of this predicted phenomenon consistently with special relativity [6–10].

In the present work we take a different attitude; given that the zitterbewegung is yet to be experimentally observed directly on a free particle, we ask the following question: is the zitterbewegung a necessary feature of the physics of a spin-one-half particle? In other words, it is asked whether Dirac's theory is the unique possible theory for this kind of particle. This question is particularly interesting in relation to the fact that there is an approach for developing the quantum theory of a free particle, an alternative to canonical

quantization, which makes use of group-theoretical methods. Differently from canonical quantization, this alternative method deductively develops the theory from two basic principles: the invariance of the theory and the covariance of the position with respect to relativistic transformations. A theory developed according to this method does not suffer the problems of canonical quantization: the validity of its predictions is implied by the validity of the basic principles (The eventual occurrence of an inconsistency would imply the failure of one of the basic principles, whose validity should be consequently questioned.), and therefore, inconsistencies, such as the negative probabilities of the Klein–Gordon theory, should be avoided. For these reasons, our work strictly complies with this epistemically grounded method.

A first important implication obtained by this alternative approach is that the Hilbert space of the quantum theory of a free particle must admit a *transformer triplet* $(U, \mathcal{S}, \mathcal{T})$ formed by a unitary representation $U$ of the universal covering $\tilde{\mathcal{P}}_+^\uparrow$ of the proper orthochronous Poincaré group $\mathcal{P}_+^\uparrow$, which realizes the transformations of quantum observables implied by the transformations of $\mathcal{P}_+^\uparrow$ in the quantum theory of the system ; $\mathcal{S}$ and $\mathcal{T}$ are operators realizing the quantum transformations implied by the space-inversion and time-reversal transformations.

Outstanding researchers, such as Wigner and Bargmann [11–13], were the pioneers of this alternative method, pursued until recent times [14–18]. However, these approaches are not adequate to deal with Dirac's theory, because they exclude transformer triplets with $\mathcal{S}$ antiunitary or with $\mathcal{T}$ unitary; instead, as argued in remark 4.1, the triplet of a quantum theory for a free Dirac particle, i.e., for an elementary free particle with positive mass, spin one-half and four-component wave functions, must have $\mathcal{S}$ antiunitary or $\mathcal{T}$ unitary. In the present work, the quantum theories of interest are deductively derived from the basic principles without a priori preclusions about the features of $\mathcal{S}$ or $\mathcal{T}$.

Our results ascertain that Dirac's theory is just one particular theory for a free Dirac particle. The class of possible quantum theories for free Dirac particle is explicitly determined. A special subclass is identified whose theories are free from singular features, such as luminal velocity for massive particles and the zitterbewegung. Each theory is characterized by a particular transformation property of position with respect to boosts. Dirac's theory is *completely* characterized by such a peculiar transformation property.

A necessary condition of validity for every one of the possible theories is that the experimentally successful predictions include all the predictions of Dirac's theory confirmed by experiment. The identification of *the* valid theory is attained if all its predictions are experimentally confirmed, while any different theory has at least one prediction that differs from the corresponding prediction of the first one, hence falsified by experiment. Since many predictions of Dirac's theory refer to an interacting particle, the comparison of the predictions requires the availability of possible alternative theories for interacting particle derived according to our approach. These nonfree-particle alternative theories are yet to be accomplished [19], so for the time being, the comparison can be carried out only among the predictions of different free-particle theories. We identify an ideal experimental test for a free Dirac particle, such that the predictions of the results by Dirac's theory and one of the alternative theories are different. In Section 2 the concept of an isolated system is introduced as a system whose physical theory satisfies the invariance principle under relativistic transformations. Then, it is shown how this principle implies that the Hilbert space of the quantum theory of the system must contain a transformer triplet $(U, \mathcal{S}, \mathcal{T})$ realizing the quantum transformations.

Section 3 specializes to a particular kind of isolated system, the elementary free particle, i.e., an isolated system whose quantum theory admits a unique position operator $\mathbf{Q}$ satisfying the covariance properties of the position with respect to relativistic transformations, such that the system $(U, \mathcal{S}, \mathcal{T}; \mathbf{Q})$ is irreducible. The classes of possible theories are identified.

Section 4 synthetically formulates Dirac's theory and the implied concept of a Dirac particle is introduced. The singular predictions of luminal velocity for massive particles

and of the zitterbewegung are highlighted. However, it is also proven that according to the present approach the class of possible theories for Dirac particles contains much more theories than Dirac's one.

Within the already identified class of theories able to describe a free Dirac particle, Section 5 completely identifies the special subclass characterized by a "diagonal" position operator. It is proven that for all theories of this subclass the singular features of Dirac's theory are absent.

Different theories of elementary free particles predict different transformation properties of position with respect to boosts, in general. Section 6 argues that within the class able to describe free Dirac particles, Dirac's theory is completely characterized by its peculiar transformation property.

Section 7 argues that the experimental verification of the validity of the theories could be implemented, at least in principle.

## 2. General Implications of Poincaré Invariance

After fixing the notation and the basic prerequisites in Section 2.1, in this section, we show that the principle of invariance with respect to Poincaré transformations is a sufficient condition for identifying the general structure of the quantum theory of any isolated system; in particular, this principle implies that the Hilbert space of the quantum theory of an isolated system must contain

- A unitary representation of the universal covering group $\tilde{\mathcal{P}}_+^\uparrow$ of the proper orthochronous Poincaré group $\mathcal{P}_+^\uparrow$ that realizes the quantum transformations of quantum observables, implied by transformations in $\mathcal{P}_+^\uparrow$;
- Two operators $\mathfrak{T}$ and $\mathfrak{S}$ that realize the quantum transformations implied by time reversal and space inversion.

The operators of the system $(U(\tilde{\mathcal{P}}_+^\uparrow), \mathfrak{S}, \mathfrak{T})$ so obtained must be related by the specific constraints shown in Section 2.3, implied by the specific group's structural properties.

### 2.1. Basic Prerequisites

2.1.1. The Formalism of a Quantum Theory

The basic mathematical structures of the general formalism of the quantum theory of a physical system based on a Hilbert space $\mathcal{H}$ are

- The set $\Omega(\mathcal{H})$ of all self-adjoint operators representing observables;
- The set $\mathcal{S}(\mathcal{H})$ of all density operators $\rho$ identified with quantum states; a quantum state $\rho$ is pure if and only if it is a one-dimensional projection operator, i.e., if $\rho = |\psi\rangle\langle\psi|$, where $\psi \in \mathcal{H}$ and $\|\psi\| = 1$; in this case, $\psi$ is called the state vector of the system;
- The group $\mathcal{U}(\mathcal{H})$ of all unitary operators;
- The larger set $\mathcal{V}(\mathcal{H})$ of all unitary or antiunitary operators.

For the present work, a very important mathematical concept is the Poincaré group $\mathcal{P}$, because it is the symmetry group of a free particle.

2.1.2. Poincaré Group

The Poincaré group $\mathcal{P}$ is the group generated by $\{\mathcal{P}_+^\uparrow, \mathfrak{t}, \mathfrak{s}\}$, where $\mathcal{P}_+^\uparrow$ is the proper orthochronous Poincaré group, and by $\mathfrak{t}$ and $\mathfrak{s}$, which are the time-reversal and space-inversion transformations.

The proper orthochronous group $\mathcal{P}_+^\uparrow$ is a connected group generated by 10 one-parameter subgroups, namely, the subgroup $\mathcal{T}_0$ of time translations, the three subgroups $\mathcal{T}_j$ ($j = 1, 2, 3$) of spatial translations, the three subgroups $\mathcal{R}_j$ of spatial rotations and the three subgroups $\mathcal{B}_j$ of Lorentz boosts, relative to the three spatial axes $x_j$. Time reversal $\mathfrak{t}$ and space inversion $\mathfrak{s}$ are not connected with the identity transformation $e \in \mathcal{P}$. Given any vector $\underline{x} = (x_0, \mathbf{x}) \in \mathbb{R}^4$, where $x_0$ is called the *time component* of $\underline{x}$ and $\mathbf{x} = (x_1, x_2, x_3)$ is called the *spatial component* of $\underline{x}$, time reversal $\mathfrak{t}$ transforms $\underline{x} = (x_0, \mathbf{x})$ into $(-x_0, \mathbf{x})$ and space inversion $\mathfrak{s}$ transforms $\underline{x} = (x_0, \mathbf{x})$ into $(x_0, -\mathbf{x})$.

The universal covering group of $\mathcal{P}_+^\uparrow$ is the semidirect product $\tilde{\mathcal{P}}_+^\uparrow = \mathbb{R}^4 \circledS SL(2,\mathbb{C})$ of the time–space translation group $\mathbb{R}^4$ and the group $SL(2,\mathbb{C}) = \{\underline{\Lambda} \in GL(2,\mathbb{C}) \mid \det \underline{\Lambda} = 1\}$. Accordingly, $\tilde{\mathcal{P}}_+^\uparrow$ is simply connected and there is a canonical homomorphism h : $\tilde{\mathcal{P}}_+^\uparrow \to \mathcal{P}_+^\uparrow$, $\tilde{g} \to \mathsf{h}(\tilde{g}) \in \mathcal{P}_+^\uparrow$, which becomes an isomorphism when restricted to small enough neighbourhoods of the identity $(0, \mathbb{1}_{\mathbb{C}^2})$ of $\tilde{\mathcal{P}}_+^\uparrow$. Within $\tilde{\mathcal{P}}_+^\uparrow$, the subgroups that correspond to the subgroups $\mathcal{T}_0, \mathcal{T}_j, \mathcal{R}_j, \mathcal{B}_j, \mathcal{L}_+^\uparrow$ of $\mathcal{P}_+^\uparrow$, respectively, through the homomorphism h, are denoted by $\tilde{\mathcal{T}}_0, \tilde{\mathcal{T}}_j, \tilde{\mathcal{R}}_j, \tilde{\mathcal{B}}_j$ and $\tilde{\mathcal{L}}_+^\uparrow$; they are all one-parameter additive subgroups; to be precise, $\tilde{\mathcal{B}}_j$ is additive with respect to the parameter $\varphi(u) = \frac{1}{2} \ln \frac{1+u}{1-u}$, not with respect to the *relative velocity u*.

### 2.2. Derivation of the Theory of Isolated System

A free particle is a particular kind of isolated system. Therefore, it is worth beginning with the derivation of the general structure of the quantum theory of an isolated system.

By $\mathcal{F}$, we denote the class of the (inertial) reference frames that move uniformly with respect to each other; if $\Sigma$ belongs to $\mathcal{F}$, then $\Sigma_g$ denotes the frame related to $\Sigma$ by such $g$, for every $g \in \mathcal{P}$. An *isolated system* is a system for which the following *invariance principle* holds.

$\mathcal{IP}$    *The theory of an isolated system is invariant with respect to changes of frames within the class $\mathcal{F}$.*

Given an observable represented by a self-adjoint operator $A$, let $\mathcal{M}_A$ be a procedure to perform the measurement of this observable; then the invariance principle implies that another measuring procedure $\mathcal{M}'_A$ must exist, which is with respect to $\Sigma_g$ identical to what $\mathcal{M}_A$ is with respect to $\Sigma$, otherwise, the principle $\mathcal{IP}$ would be violated. Hence, $\mathcal{IP}$ implies the existence of a mapping

$$S_g : \Omega(\mathcal{H}) \to \Omega(\mathcal{H}), \quad A \to S_g[A],$$

where $S_g[A]$ is the self-adjoint operator that represents the observable measured by $\mathcal{M}'_A$. The mapping $S_g$ is called the *quantum transformation associated to g* [19].

To every element $\tilde{g}$ of the covering group $\tilde{\mathcal{P}}_+^\uparrow$, we can associate the quantum transformation $S_{\mathsf{h}(\tilde{g})} \equiv S_{\tilde{g}}$ through the canonical homomorphism h. By making use of group-theoretical methods, in [19] it is proven that

- A continuous unitary representation $U$ of $\tilde{\mathcal{P}}_+^\uparrow$ exists such that $S_{\tilde{g}}[A] = U_{\tilde{g}} A U_{\tilde{g}}^{-1}$.
- Two operators $\mathcal{S}$ and $\mathcal{T}$, each of them unitary or antiunitary, exist such that (The unitarity of $\mathcal{S}$ and $\mathcal{T}$ cannot be proven in general because $\mathfrak{s}$ and $\mathfrak{t}$ ar not connected with the identity element of $\mathcal{P}$ [19].)
  $S_{\mathfrak{s}}[A] = \mathcal{S}A\mathcal{S}^{-1}$ and $T_{\mathfrak{s}}[A] = \mathcal{T}A\mathcal{T}^{-1}$.

Thus, the principle $\mathcal{IP}$ has the following fundamental implication.

$(\mathcal{FI})$    *In the quantum theory of an isolated system, a triplet $(U, \mathcal{S}, \mathcal{T})$ must exist, called the transformer triplet of the theory, formed by a continuous representation $U$ of $\tilde{\mathcal{P}}_+^\uparrow$ and by two operators $\mathcal{S}, \mathcal{T} \in \mathcal{V}(\mathcal{H})$ such that*

$$S_{\tilde{g}}[A] = U_{\tilde{g}} A U_{\tilde{g}}^{-1}, \quad \mathcal{S}A\mathcal{S}^{-1} = S_{\mathfrak{s}}[A], \quad \mathcal{T}A\mathcal{T}^{-1} = S_{\mathfrak{t}}[A], \text{for all } A \in \Omega(\mathcal{H}). \quad (1)$$

### 2.3. Constraints for the Transformer Triplet

According to Stone's theorem [19,20], the representation $U$ of the transformer triplet $(U, \mathcal{S}, \mathcal{T})$ of the quantum theory of an isolated system is completely determined by ten self-adjoint operators $P_0, P_j, J_j$ and $K_j$, called self-adjoint generators, such that if $\tilde{g}$ is an

element of the one-parameter subgroup $\tilde{\mathcal{T}}_0$ (resp., $\tilde{\mathcal{T}}_j$, $\tilde{\mathcal{R}}_j$ and $\tilde{\mathcal{B}}_j$) identified by the value $t$ (resp., $a$, $\theta$ and $u$) of the parameter, then

$$U_{\tilde{g}} = e^{iP_0 t}, \quad (\text{resp., } U_{\tilde{g}} = e^{iP_j a}, \ U_{\tilde{g}} = e^{J_j \theta}, \ U_{\tilde{g}} = e^{iK_j \varphi(u)}. \tag{2}$$

The generator $P_0$ relative to time translations rules over quantum dynamics is according to the equations

$$(i) \quad \dot{A}_t = i[P_0, A_t], \qquad (ii) \quad i\frac{\partial \psi_t}{\partial t} = P_0 \psi_t. \tag{3}$$

The group's structural properties of $\mathcal{P}$ imply that the self-adjoint generators and the time-reversal and space-inversion operators $\mathcal{S}$ and $\mathcal{T}$ can always be chosen so that relations (4)–(9) hold [11,14].

$$
\begin{array}{lll}
(i) \quad [P_j, P_k] = \mathbb{O}, & (ii) \ [J_j, P_k] = i\hat{\epsilon}_{jkl} P_l, & (iii) \ [J_j, J_k] = i\hat{\epsilon}_{jkl} J_l, \\
(iv) \quad [J_j, K_k] = i\hat{\epsilon}_{jkl} K_l, & (v) \ [K_j, K_k] = -i\hat{\epsilon}_{j,k,l} J_l, & (vi) \ [K_j, P_k] = i\delta_{jk} P_0, \\
(vii) \ [P_j, P_0] = \mathbb{O}, & (viii) \ [J_j, P_0] = \mathbb{O}, & (ix) \ [K_j, P_0] = iP_j,
\end{array} \tag{4}
$$

where $\hat{\epsilon}_{jkl}$ is the Levi-Civita symbol $\epsilon_{jkl}$ restricted by the condition $j \neq l \neq k$.

If $\mathcal{S}$ *is unitary*, then its arbitrary phase factor can be chosen so that $\mathcal{S}^2 = \mathbb{1}$, and
$$[\mathcal{S}, P_0] = \mathbb{O}, \quad \mathcal{S}P_j = -P_j \mathcal{S}, \quad [\mathcal{S}, J_k] = \mathbb{O}, \quad \mathcal{S}K_j = -K_j \mathcal{S}; \tag{5}$$

If $\mathcal{S}$ is antiunitary, then $\mathcal{S}^2 = c\mathbb{1}$, so $\mathcal{S}^{-1} = c\mathcal{S}$, where $c = 1$ or $c = -1$, and
$$\mathcal{S}P_0 = -P_0 \mathcal{S}, \quad [\mathcal{S}, P_j] = \mathbb{O}, \quad \mathcal{S}J_k = -J_k \mathcal{S}, \quad \mathcal{S}K_j = K_j \mathcal{S}; \tag{6}$$

If $\mathcal{T}$ is unitary, then its phase factor can be chosen so that $\mathcal{T}^2 = \mathbb{1}$, and
$$\mathcal{T}P_0 = -P_0 \mathcal{T}, \quad [\mathcal{T}, P_j] = \mathbb{O}, \quad [\mathcal{T}, J_k] = \mathbb{O}, \quad \mathcal{T}K_j = -K_j \mathcal{T}; \tag{7}$$

If $\mathcal{T}$ is antiunitary, then $\mathcal{T}^2 = c\mathbb{1}$, so $\mathcal{T}^{-1} = c\mathcal{T}$, either $c = 1$ or $c = -1$, and
$$\mathcal{T}P_0 = P_0 \mathcal{T}, \quad \mathcal{T}P_j = -P_j \mathcal{T}, \quad \mathcal{T}J_k = -J_k \mathcal{T}, \quad \mathcal{T}K_j = K_j \mathcal{T}; \tag{8}$$

$$\mathcal{S}\mathcal{T} = \omega \mathcal{T}\mathcal{S}, \quad \text{with } \omega \in \mathbb{C} \text{ and } |\omega| = 1 \tag{9}$$

The *helicity* operator is defined by $\hat{\lambda} = \frac{\mathbf{J} \cdot \mathbf{P}}{P}$, where $P = \sqrt{P_1^2 + P_2^2 + P_3^2}$. The following relation

$$\mathcal{S}\hat{\lambda}\mathcal{S}^{-1} = -\hat{\lambda} \tag{10}$$

is implied by (5) but also by (6). Therefore, it holds independently of the unitary or antiunitary character of $\mathcal{S}$.

By making use of (4)–(8), it can be proven that the following relations hold.

$$[V, P_0^2 - \mathbf{P}^2] = \mathbb{O}, \quad [V, W_0^2 - \mathbf{W}^2] = \mathbb{O}, \quad \text{for all } V \in U(\tilde{\mathcal{P}}_+^{\uparrow}) \cup \{\mathcal{T}, \mathcal{S}\}, \tag{11}$$

where $W_0 = \mathbf{J} \cdot \mathbf{P}$ and $\mathbf{W} = P_0 \mathbf{J} + \mathbf{P} \wedge \mathbf{K}$ form the *Pauli–Lubański four-operator* $(W_0, \mathbf{W})$.

## 3. Theories of Elementary Free Particles

This section derives the structures of the possible quantum theories of a particular kind of isolated system, the *elementary free particle*. Following [19], it is shown how the triplets of the quantum theories of an elementary free particle can be classified with respect to a *mass* parameter $\mu$ and a *spin* parameter $s$. The classes of theories of interest are then identified.

A *free particle* is an isolated system whose quantum theory is endowed with a unique *position observable*, that is to say with a *unique* three-operator $\mathbf{Q} \equiv (Q_1, Q_2, Q_3)$, with all $Q_j$ self-adjoint, satisfying the following conditions

($Q$.1) $[Q_j, Q_k] = \mathbb{O}$, for all $j, k = 1, 2, 3$. This condition establishes that a measurement of position yields all three values of the coordinates of the same specimen of the system.

(Q.2) For every $g \in \mathcal{P}$, the three-operator $(Q_1, Q_2, Q_3) \equiv \mathbf{Q}$ and the transformed position operator $S_g[\mathbf{Q}] \equiv (S_g[\mathbf{Q}_1], S_g[\mathbf{Q}_2], S_g[\mathbf{Q}_3])$ satisfy the specific relations implied by the transformation properties of position with respect to $g$.

An *elementary* free particle is a free particle for which the system $(U(\tilde{\mathcal{P}}_+^\uparrow), \mathcal{S}, {}^\triangleleft\!T; \mathbf{Q})$ is an irreducible system of operators. As proven in [19], the transformer triplet $(U, \mathcal{S}, {}^\triangleleft\!T)$ of the quantum theory of an elementary free particle must be *irreducible*. Thus, the identification of all possible theories of an elementary free particle can be carried out in two steps: first by identifying all irreducible triplets $(U, \mathcal{S}, {}^\triangleleft\!T)$, i.e., by (2), all irreducible operator systems $(P_0, P_j, J_j, K_j, \mathcal{S}, {}^\triangleleft\!T)$, $j = 1, 2, 3$, such that (4)–(9) hold, and then selecting those systems for which a unique position operator $\mathbf{Q}$ exists.

### 3.1. The First Step for Positive-Mass Elementary Particle

If the triplet $(U, \mathcal{S}, {}^\triangleleft\!T)$ is irreducible, then, by a straightforward application of Schur's lemma, from (11), we imply that the quantum theory of an elementary free particle is characterized by two numbers $\mu \in \mathbb{C}$, $\varpi \in \mathbb{R}$, with $\mu^2 \in \mathbb{R}$, such that

$$P_0^2 - \mathbf{P}^2 = \mu^2 \mathbb{I}, \qquad W^2 \equiv W_0^2 - (W_1^2 + W_2^2 + W_3^2) = \varpi \mathbb{I}. \tag{12}$$

Parameter $\mu$ is called *mass*. The present work is interested with positive-mass particle theories, i.e., with triplets with $\mu > 0$. For these triplets, the relation $\varpi = -\mu^2 s(s+1)$ holds, where the value $s$ is called *spin* and it is an integral or half-integral number: $s \in \frac{1}{2}\mathbb{N}$. For any pair $(\mu, s) \in \mathbb{R}_+ \times \frac{1}{2}\mathbb{N}$, there is at least an irreducible triplet.

It can be proven that for every triplet with mass $\mu$, the spectrum $\sigma(P_0)$ has three mutually exclusive possibilities:

$\sigma(P_0) = (-\infty, -\mu]$ ;
$\sigma(P_0) = [\mu, \infty)$ ;
$\sigma(P_0) = (-\infty, -\mu] \cup [\mu, \infty)$.

In particular, the following characterization holds [19].

**Theorem 1.** $\sigma(P_0) = (-\infty, -\mu] \cup [\mu, \infty)$ *if and only if* ${}^\triangleleft\!T$ *is unitary or* $\mathcal{S}$ *is antiunitary.*

Therefore, denoting by $\mathcal{I}(\mu, s)$ the class of all positive mass irreducible triplets with parameters $(\mu, s)$, we have

$$\mathcal{I}(\mu, s) = \mathcal{I}^+(\mu, s) \cup \mathcal{I}^-(\mu, s) \cup \mathcal{I}^{-+}(\mu, s). \tag{13}$$

where $\mathcal{I}^-(\mu, s)$ (resp., $\mathcal{I}^+(\mu, s)$, $\mathcal{I}^{-+}(\mu, s)$) is the class of irreducible triplets with $\sigma(P_0) = (-\infty, -\mu]$ (resp., $\sigma(P_0) = [\mu, \infty)$, $\sigma(P_0) = (-\infty, -\mu] \cup [\mu, \infty)$).

3.1.1. The Classes $\mathcal{I}^+(\mu, s)$ and $\mathcal{I}^-(\mu, s)$

The representation $U$ of a triplet in $\mathcal{I}^+(\mu, s)$ can be irreducible or not. With modulo unitary isomorphisms, fixing $(\mu, s)$, there is only one triplet $(U, \mathcal{S}, {}^\triangleleft\!T)$ in $\mathcal{I}^+(\mu, s)$ with $U$ irreducible, identified by the following Hilbert space and operators:

– The Hilbert space is $\mathcal{H} = L_2(\mathbb{R}^3, \mathbb{C}^{2s+1}, d\nu)$, where $d\nu(\underline{p}) = \frac{dp_1 dp_2 dp_3}{\sqrt{\mu^2 + \mathbf{p}^2}}$;

– The generators are defined by $\quad (P_j \psi)(\mathbf{p}) = p_j \psi(\mathbf{p}), \quad (P_0 \psi)(\underline{p}) = p_0 \psi(\underline{p})$,
  $\mathsf{J}_k = \mathsf{J}_k^{(0)} + S_k$ and $K_j = i p_0 \frac{\partial}{\partial p_j} - \frac{(\mathbf{S} \wedge \mathbf{p})_j}{\mu + p_0}$, with $\mathsf{J}_k^{(0)} = -i\left( p_l \frac{\partial}{\partial p_j} - p_j \frac{\partial}{\partial p_l} \right)$; $\qquad$ (14)

– The space-inversion and time-reversal operators are $\mathcal{S} = \mathsf{Y}$, $\quad {}^\triangleleft\!T = \tau \mathcal{K} \mathsf{Y}$,

where $p_0 = \sqrt{\mu^2 + \mathbf{p}^2}$, $S_1, S_2, S_3$ are the spin operators of $\mathbb{C}^{2s+1}$, $\tau \in \mathcal{U}(\mathbb{C}^{2s+1})$ is a matrix such that $\tau \bar{S}_j \tau^{-1} = -S_j$, $\mathcal{K}$ and $\mathsf{Y}$ are defined by $\mathcal{K}\psi(\mathbf{p}) = \overline{\psi(\mathbf{p})}$ and $(\mathsf{Y}\psi)(\mathbf{p}) = \psi(-\mathbf{p})$.

Analogously, there is only one irreducible triplet in $\mathcal{I}^-(\mu, s)$ with $U$ irreducible, It differs from that of $\mathcal{I}^+(\mu, s)$ by $P_0 = -p_0$ and $K_j = -\mathsf{K}_j$.

Moreover, irreducible triplets in $\mathcal{I}^{\pm}(\mu, s)$ with $U$ *reducible* exist. They were identified in [19].

### 3.1.2. The Class $\mathcal{I}^{-+}(\mu, s)$

The representation $U$ of a triplet in $\mathcal{I}^{-+}(\mu, s)$ turns out to be always reducible, namely, $U = U^+ \oplus U^-$ where $U^{\pm}$ belongs to a triplet in $\mathcal{I}^{\pm}(\mu, s)$. Moreover, $U^+$ is irreducible if and only if $U^-$ is irreducible.

Fixing $(\mu, s)$, there are six inequivalent triplets in $\mathcal{I}^{-+}(\mu, s)$ with $U^+$, and hence $U^-$, irreducible. All these triplets have the same Hilbert space and the same generators:

$$\mathcal{H} = L_2(\mathbb{R}^3, \mathbb{C}^{2s+1}, d\nu) \oplus L_2(\mathbb{R}^3, \mathbb{C}^{2s+1}, d\nu);$$

if $\psi \in \mathcal{H}$, then $\psi \equiv \psi_1 \oplus \psi_2 \equiv \begin{bmatrix} \psi_1 \\ \psi_2 \end{bmatrix}$, $\psi_1, \psi_2 \in L_2(\mathbb{R}^3, \mathbb{C}^{2s+1}, d\nu))$.

$$P_j = \begin{bmatrix} p_j & 0 \\ 0 & p_j \end{bmatrix}, P_0 = \begin{bmatrix} p_0 & 0 \\ 0 & -p_0 \end{bmatrix}, J_k = \begin{bmatrix} J_k & 0 \\ 0 & J_k \end{bmatrix}, K_j = \begin{bmatrix} K_j & 0 \\ 0 & -K_j \end{bmatrix}. \tag{15}$$

The six triplets $(U, \mathcal{S}_n, {}^{\P}\mathsf{T}_n)$, $n = 1, 2, \ldots, 6$, differ for the different pairs $(\mathcal{S}_n, {}^{\P}\mathsf{T}_n)$:

$$\begin{aligned} {}^{\P}\mathsf{T}_1 &= \tau \mathcal{K} \mathsf{Y} \begin{bmatrix} 1 & 0 \\ 0 & 1 \end{bmatrix}, & \mathcal{S}_1 &= \begin{bmatrix} 0 & \tau \\ \tau & 0 \end{bmatrix} \mathcal{K}; & {}^{\P}\mathsf{T}_2 &= \tau \mathcal{K} \mathsf{Y} \begin{bmatrix} 1 & 0 \\ 0 & 1 \end{bmatrix}, & \mathcal{S}_2 &= \begin{bmatrix} 0 & \tau \\ -\tau & 0 \end{bmatrix} \mathcal{K}; \\[2mm] {}^{\P}\mathsf{T}_3 &= \begin{bmatrix} 0 & 1 \\ 1 & 0 \end{bmatrix}, & \mathcal{S}_3 &= \begin{bmatrix} 0 & \tau \\ \tau & 0 \end{bmatrix} \mathcal{K}; & {}^{\P}\mathsf{T}_4 &= \begin{bmatrix} 0 & 1 \\ 1 & 0 \end{bmatrix}; & \mathcal{S}_4 &= \begin{bmatrix} 0 & \tau \\ -\tau & 0 \end{bmatrix} \mathcal{K}; \\[2mm] {}^{\P}\mathsf{T}_5 &= \begin{bmatrix} 0 & 1 \\ 1 & 0 \end{bmatrix}, & ; \mathcal{S}_5 &= \mathsf{Y} \begin{bmatrix} 1 & 0 \\ 0 & 1 \end{bmatrix}; & {}^{\P}\mathsf{T}_6 &= \begin{bmatrix} 0 & 1 \\ 1 & 0 \end{bmatrix}, & \mathcal{S}_6 &= \mathsf{Y} \begin{bmatrix} 1 & 0 \\ 0 & -1 \end{bmatrix}. \end{aligned} \tag{16}$$

The search for irreducible triplets in $\mathcal{I}^{-+}(\mu, s)$ with $U^+$, and hence $U^-$, *reducible* is not of interest in the present work; in fact, we shall see that Dirac's theory is based on triplets with $U^+$ irreducible. A partial account was given in [19].

### 3.2. Second Step

In order to complete the identification of the theories of an elementary free particle, the transformer triplets that admit a unique commutative three-operator $\mathbf{Q} = (Q_1, Q_2, Q_3)$ for which $(Q.2)$ holds must be selected. Condition $(Q.2)$ for spatial translations and rotations, i.e., for $U_{\tilde{g}} = e^{-iP_j a}$ and $U_{\tilde{g}} = e^{-iJ_j \theta}$, holds if and only if the following commutation relations hold [19]:

$$[Q_k, P_j] = i\delta_{jk} \quad \text{and} \quad [J_j, Q_k] = i\hat{\epsilon}_{jkl} Q_l. \tag{17}$$

For space inversion $\mathcal{s}$ and time reversal $\mathcal{t}$, condition $(Q.2)$ is $S_{\mathcal{t}}[\mathbf{Q}] = {}^{\P}\mathsf{T}\mathbf{Q}{}^{\P}\mathsf{T}^{-1} = \mathbf{Q}$ and $S_{\mathcal{s}}[\mathbf{Q}] = \mathcal{s}\mathbf{Q}\mathcal{s}^{-1} = -\mathbf{Q}$, i.e.,

$$(i) \quad {}^{\P}\mathsf{T}\mathbf{Q} = \mathbf{Q}{}^{\P}\mathsf{T} \quad \text{and} \quad (ii) \quad \mathcal{s}\mathbf{Q} = -\mathbf{Q}\mathcal{s}. \tag{18}$$

Relations (17) and (18) are general conditions to be satisfied in any theory of elementary free particle.

The next step should be to determine the conditions implied by the transformation properties of the position with respect to boosts, i.e., the relations for $[K_j, Q_k]$. Unfortunately, these relations are not available. Let $g = \mathsf{h}(\tilde{g})$ be a boost characterized by a relative velocity $\mathbf{u} = (u, 0, 0)$. The present concept of quantum transformation entails that the explicit relation between $S_{\tilde{g}}[\mathbf{Q}]$ and $\mathbf{Q}$ must identify, if $\mathbf{x}$ is the position outcome at time $t = 0$ with respect to $\Sigma$, the corresponding position $\mathbf{y_x}$ with respect to $\Sigma_g$ but at time $t' = 0$ *with respect to* $\Sigma_g$. Special relativity does not provide such a correspondence. Indeed, if the outcome of position is $\mathbf{x} = (x_1, x_2, x_3)$ at time $t = 0$ in $\Sigma$, then according to Lorentz's transformations the position with respect to $\Sigma_g$ is $\mathbf{y} = \left( \frac{x_1}{\sqrt{1-u^2}}, x_2, x_3 \right)$, but at time $t' = \frac{-ux_1}{\sqrt{1-u^2}}$, not at $t' = 0$!

An attempt to solve the problem could be to introduce a *four-position* $\underline{Q} = (Q_0, \mathbf{Q})$, where $Q_0$ is just the *time* quantum observable, i.e., the time when the measurement of the spatial coordinates represented by $\mathbf{Q}$ takes place. Then, special relativity would imply the following transformation properties of the four-position with respect to boosts: $S_g[Q_0] = \frac{Q_0 - uQ_1}{\sqrt{1-u^2}}$, $S_g[Q_1] = \frac{Q_1 - uQ_0}{\sqrt{1-u^2}}$, $S_g[Q_2] = Q_2$, $S_g[Q_3] = Q_3$. However, in any quantum theory of localizable particles, *such a time cannot be a quantum observable* [19].

Therefore, the second step can be carried out only partially, by selecting the triplets that admit a solution $\mathbf{Q}$ of (17) and (18). In so doing, it has been found [18,19] that for *zero spin*, i.e., within the classes of triplets with $s = 0$, there are subclasses that admit a *unique* solution of (17) and (18). Namely, every triplet in $\mathcal{I}^+(\mu, 0)$ and every triplet in $\mathcal{I}^-(\mu, 0)$ have a unique solution $\mathbf{Q} = \mathbf{F}$, where

$$F_j = i\frac{\partial}{\partial p_j} - \frac{i}{2p_0^2}p_j \tag{19}$$

are the Newton–Wigner operators [21]. In the class $\mathcal{I}^{-+}(\mu, 0)$, two inequivalent triplets have been identified [19] for which there is a unique solution $\mathbf{Q} = \begin{bmatrix} \mathbf{F} & 0 \\ 0 & \mathbf{F} \end{bmatrix}$ of (17) and (18). These particular triplets give rise to possible theories of elementary free particles with zero spin. In each such theory, the transformation properties of the position with respect to boosts are determined by simply computing the commutator $[K_j, Q_k]$. In all cases, this computation yields

$$[K_j, Q_k] = -\frac{i}{2}(Q_j\dot{Q}_k + \dot{Q}_k Q_j). \tag{20}$$

The uniqueness of solutions of (17) and (18) always fails for triplets with $s > 0$, i.e., for nonzero spin. In general, different solutions give rise to different relations for the commutators $[K_j, Q_k]$; they correspond to different transformation properties of the position with respect to boosts.

If any of these properties is valid, this is not a question that can be settled on a purely theoretical ground.

## 4. Dirac's Theory

In this section Dirac's theory with its peculiar and singular features is outlined. In particular, the class of theories of our approach is identified, where the theory of a free *Dirac particle* must be formulated.

### 4.1. Dirac Theory and Zitterbewegung

Let us synthetically formulate Dirac's theory [1,2] for spin-one-half free particle.

– The state vectors are four-component complex functions on $\mathbb{R}^3$: $\varphi(\mathbf{x}) = \begin{bmatrix} \varphi^{(1)}(\mathbf{x}) \\ \varphi^{(2)}(\mathbf{x}) \end{bmatrix}$, with $\varphi^{(n)}(\mathbf{x}) \in L_2(\mathbb{R}^3, \mathbb{C}^2)$.
– The position operator $\mathbf{Q}^D$ is defined by $Q_j^D \varphi(\mathbf{x}) = x_j \varphi(\mathbf{x})$, i.e., $\mathbf{Q}^D = \mathbf{x}$;
– The self-adjoint generators relative to spatial translations are $P_j^D = -i\frac{\partial}{\partial x_j}$;
– The operator $P_0^D = \mu\beta + \alpha_1 P_1^D + \alpha_2 P_2^D + \alpha_3 P_3^D \equiv (\mu\beta + \boldsymbol{\alpha} \cdot \mathbf{P}^D)$ is the self-adjoint generator relative to time translation, where $\beta$ and $\alpha_j$ are the Dirac matrices [2].

Hence, denoting the state vector of the system at time $t$ by $\varphi_t$, the dynamical equation of Dirac's theory is *Dirac's equation*:

$$i\frac{\partial \varphi_t}{\partial t} = (\mu\beta + \boldsymbol{\alpha} \cdot \mathbf{P}^D)\varphi_t. \tag{21}$$

Perhaps the most important result of Dirac's theory is the prediction of the existence of antiparticles. However, Dirac's theory encountered several criticisms, reviewed for instance in [3]. A particularly debated feature is the fact that if any component of the "velocity" operator is computed according to the general law (3) of quantum theory, i.e., if $\dot{Q}_j^D = i[P_0^D, Q_j^D]$ is computed, then the result is $\dot{Q}_j^D = -\alpha_j$; since $\alpha_j^2 = 1$ we have to conclude that the possible value of this velocity is $\pm 1$, which in the present treatment is the speed of light.

This singular prediction of Dirac's theory received great attention in the scientific literature [4–8]. A lot of papers were devoted to recovering consistency with special relativity, at the cost of assigning the particle a very complicated kinematic behaviour [5,9,10]. The detailed study [4,5] of the behaviour of solutions of Dirac's equation proved that the position operator at time $t$ turns out to be

$$\mathbf{Q}^{(t)} = \mathbf{A}_0 + \frac{\mathbf{P}^D}{P_0^D}t + \frac{i}{2P_0^D}\mathbf{B}_0 e^{2iP_0^D t}, \tag{22}$$

where $\mathbf{A}_0$ and $\mathbf{B}_0$ are constant three-operators. This complicated oscillatory behaviour has been called the *zitterbewegung*. The problem addressed in the present work is different from that faced by these studies. Our aim is to identify quantum theories of nonzero-spin particles, which can be consistently derived from the physical principles specifying an elementary free particle and which are not affected by singular predictions such as the zitterbewegung.

*4.2. The Class of Dirac's Theory in the Present Approach*

Dirac proved that his theory for spin-one-half particles [1,2] turns out to be consistent with Poincaré invariance; moreover, the system of self-adjoint generators is irreducible. Therefore, Dirac's theory must be unitarily equivalent to one particular solution of (17) and (18) in a triplet with $s = 1/2$. Thus, the triplet must belong to $\mathcal{I}^{-+}(\mu, 1/2)$, to $\mathcal{I}^{+}(\mu, 1/2)$ or to $\mathcal{I}^{-}(\mu, 1/2)$.

According to the results of Foldy and Wouthuysen [22] and Jordan and Mukunda [23], there is a unitary isomorphism that allows one to equivalently reformulate Dirac's theory in such a way that in the reformulated version, the state vectors are four-component functions $\psi(\mathbf{p}) = \begin{bmatrix} \psi_1(\mathbf{p}) \\ \psi_2(\mathbf{p}) \end{bmatrix}$, with $\psi_n \in L_2(\mathbb{R}^3, \mathbb{C}^2, dv)$, and the self-adjoint generators take the *canonical* form (15) with $s = 1/2$.

Thus, the genuine Dirac theory must be based on triplets in $\mathcal{I}^{-+}(\mu, 1/2)$.

**Definition 1.** *A free Dirac particle is an elementary free particle whose quantum theory is based on a triplet of $\mathcal{I}^{-+}(\mu, 1/2)$.*

**Remark 1.** *The pioneers of our method of developing the theory deductively from invariance and covariance principles were outstanding researchers, such as Wigner, Bargmann, Wightmann, Mackey and Jauch [11–13,15,17]. However, these approaches turned out to be unable to encompass theories based on triplets of $\mathcal{I}^{-+}(\mu, s)$, because they rejected the triplets with $^{\triangleleft}T$ unitary or with $_{\triangleleft}S$ antiunitary, i.e., by (16), all triplets of $\mathcal{I}^{-+}(\mu, s)$, and therefore, these approaches were not adequate to develop Dirac's theory. The reason put forward for such an exclusion was the fact that $^{\triangleleft}T$ unitary or $_{\triangleleft}S$ antiunitary implied both positive and negative values of the spectrum $\sigma(P_0)$, as implied by Theorem 1; this feature was deemed contradictory, because for a free particle, $P_0$ was identified with the kinetic energy operator, which should be positive. However, as we shall see, in quantum theories based on $\mathcal{I}^{-+}(\mu, s)$, the operator $P_0$ does not coincide with the kinetic energy operator. Thus, there is no effective motivation for ruling out unitary $^{\triangleleft}T$ and antiunitary $_{\triangleleft}S$.*

## 5. Consistent Alternatives to Dirac Theory

Dirac's theory is not the unique consistent theory of nonzero-spin particle based on triplets of $\mathcal{I}^{-+}(\mu, s)$. In Section 5.1, we explicitly identify another class of consistent theories

for Dirac particles. Then, in Section 5.2, we show that this class contains subclasses that are free from singular features.

*5.1. A Special Class of Theories*

In order to identify the theories of elementary free particle based on triplets of $\mathcal{I}^{-+}(\mu, s)$, the possible commutative position operators $\mathbf{Q}$ that satisfy (17) and (18) must be determined. The candidate $\mathbf{Q}$ for position operator can be written as $\mathbf{Q} = \begin{bmatrix} \mathbf{Q}^{(11)} & \mathbf{Q}^{(12)} \\ \mathbf{Q}^{(21)} & \mathbf{Q}^{(23)} \end{bmatrix}$, where the entries $\mathbf{Q}^{(nm)}$ are operators of $L_2(\mathbb{R}^3, \mathbb{C}^{2s+1}, d\nu)$ with $\mathbf{Q}^{(nm)} = (\mathbf{Q}^{(mn)})^*$, where $*$ denotes the adjoint operation. In general, the nondiagonal entries $\mathbf{Q}^{(12)}$, $\mathbf{Q}^{(21)}$ are not zero. Here, we investigate the special class of possible theories for which the solutions $\mathbf{Q}$ of (17) and (18) such that $\mathbf{Q}^{(12)}$ and hence $\mathbf{Q}^{(12)}$ are zero, i.e., the subclass for which $\mathbf{Q}$ is diagonal:

$$\mathbf{Q} = \begin{bmatrix} \mathbf{Q}^{(11)} & 0 \\ 0 & \mathbf{Q}^{(22)} \end{bmatrix}. \tag{23}$$

In so doing, we make use of the following constraint found out by Jordan [18].

(JC) *In the Hilbert space $L_2(\mathbb{R}^3, \mathbb{C}^{2s+1}; d\nu)$, let us consider the operators $J_j$ defined by (14). A commutative three-operator $\mathbf{R} = (R_1, R_2, R_3)$ of self-adjoint operators of $L_2(\mathbb{R}^3, \mathbb{C}^{2s+1}; d\nu)$ satisfies $[R_j, p_k] = i\delta_{jk}$ and $[J_j, R_k] = i\hat{\epsilon}_{jkl}R_l$ if and only if*

$$R_j = \eta(p_0, \hat{z})\mathbf{r} \wedge (\mathbf{r} \wedge \mathbf{S}) + \gamma(p_0, \hat{z})\mathbf{r} \wedge \mathbf{S}, \tag{24}$$

*where $\eta(p_0, \hat{z})$ and $\gamma(p_0, \hat{z})$ are self-adjoint operators of $L_2(\mathbb{R}^3, \mathbb{C}^{2s+1}, d\nu)$ functions of $p_0$ and of the "reduced" helicity $\hat{z} = \mathbf{r} \cdot \mathbf{S}$, with $\mathbf{r} = \frac{\mathbf{p}}{\sqrt{p_1^2 + p_2^2 + p_3^2}}$.*

By making use of (15), the condition (17) applied to a triplet $\mathbf{Q}$ satisfying (23) holds if and only if $[Q_j^{(nn)}, p_k] = i\delta_{jk}$ and $[J_j, Q_k^{(nn)}] = i\hat{\epsilon}_{jkl}Q_l^{(nn)}$. Therefore, by (JC), $Q_j^{(nn)} = \mathbf{F} + \eta_n(p_0, \hat{z})\mathbf{r} \wedge (\mathbf{r} \wedge \mathbf{S}) + \gamma_n(p_0, \hat{z})\mathbf{r} \wedge \mathbf{S}$, $n = 1, 2$. Thus, any solution of the kind in (23), which satisfies (17) in $\mathcal{I}^{-+}(\mu, s)$, has the form

$$\mathbf{Q} = \begin{bmatrix} \mathbf{F} + \eta_1\mathbf{r} \wedge (\mathbf{r} \wedge \mathbf{S}) + \gamma_1\mathbf{r} \wedge \mathbf{S} & 0 \\ 0 & \mathbf{F} + \eta_2\mathbf{r} \wedge (\mathbf{r} \wedge \mathbf{S}) + \gamma_2\mathbf{r} \wedge \mathbf{S} \end{bmatrix}. \tag{25}$$

Each particular solution is determined by the particular quadruple $\eta_1$, $\gamma_1$, $\eta_2$ and $\gamma_2$. Since $\mathbf{Q}$ must also be a solution of (18), we determine the relative further conditions on $\eta_1$, $\gamma_1$, $\eta_2$ and $\gamma_2$. These conditions are explicitly derived as implications of $^{\triangleleft}T_n\mathbf{Q} = \mathbf{Q}^{\triangleleft}T_n$ and $\mathscr{S}\mathbf{Q} = -\mathbf{Q}\mathscr{S}_n$ for each triplet in (16). This derivation can be easily carried out by making use of the following directly verifiable relations:

(i) $\tau\overline{S_k}\tau^{-1} = -S_k$,    (ii) $Y\mathbf{r} = -\mathbf{r}Y$;    (iii) $Y\mathbf{S} = \mathbf{S}Y$;    (iv) $Y\mathbf{F} = -\mathbf{F}Y$;

(v) $Y\eta_n(p_0, \hat{z}) = \eta_n(p_0, -\hat{z})Y$,    $Y\gamma_n(p_0, \hat{z}) = \gamma_n(p_0, -\hat{z})Y$;          (26)

(vi) $\mathcal{K}\mathbf{F} = -\mathbf{F}\mathcal{K}$;    (vii) $\tau\mathcal{K}Y\mathbf{F} = \mathbf{F}\tau\mathcal{K}Y$;    (viii) $\tau\mathcal{K}\mathbf{S} = -\mathbf{S}\tau\mathcal{K}$;

(ix) $\tau\mathcal{K}\hat{z} = -\hat{z}\tau\mathcal{K}$;    (x) $\tau\mathcal{K}\eta_n = \eta_n\tau\mathcal{K}$,    $\tau\mathcal{K}\gamma_n = \gamma_n\tau\mathcal{K}$.

In so doing, the following results are found.

**Theorem 2.** *A three-operator $\mathbf{Q}$ of the form in (25) satisfies $^{\triangleleft}T_n\mathbf{Q} = \mathbf{Q}^{\triangleleft}T_n$ and $\mathscr{S}_n\mathbf{Q} = -\mathbf{Q}\mathscr{S}_n$ if and only if*

**(i)** $\eta_1 = \eta_2 = 0$ *and* $\gamma_1 = \gamma_2 = \gamma$ *for* $n = 1, 2, 3, 4$;

**(ii)** $\eta_1 = \eta_2 = \eta$, *and* $\gamma_1 = \gamma_2 = \gamma$, *where $\eta$ and $\gamma$ are, respectively, odd and even with respect to $\hat{z}$, for $n = 5, 6$.*

*5.2. No Singular Features*

Every theory of the classes identified in Section 5.1, characterized by (23), is not affected by the singular predictions of Dirac's theory. Indeed, the "velocity" operator, according to (3) and (15), is given by

$$\dot{\mathbf{Q}} = i \begin{bmatrix} [p_0, \mathbf{F} + \eta_1 \mathbf{r} \wedge (\mathbf{r} \wedge \mathbf{S}) + \gamma_1 \mathbf{r} \wedge \mathbf{S}] & 0 \\ 0 & -[p_0, \mathbf{F} + \eta_2 \mathbf{r} \wedge (\mathbf{r} \wedge \mathbf{S}) + \gamma_2 \mathbf{r} \wedge \mathbf{S}] \end{bmatrix}. \quad (27)$$

Now, since $[p_0, \mathbf{r}] = [p_0, \mathbf{S}] = 0$, $[p_0, \hat{z}] = 0$ and $[p_0, \eta] = [p_0, \gamma] = 0$ hold as well, so that

$$\dot{\mathbf{Q}} = i \begin{bmatrix} [p_0, \mathbf{F}] & 0 \\ 0 & -[p_0, \mathbf{F}] \end{bmatrix} = \begin{bmatrix} \frac{\mathbf{p}}{p_0} & 0 \\ 0 & -\frac{\mathbf{p}}{p_0} \end{bmatrix}. \quad (28)$$

Therefore, $\dot{\mathbf{Q}}^2 = \frac{p^2}{p_0^2}$, where $p_0 = \sqrt{\mu^2 + \mathbf{p}^2}$, so $\dot{\mathbf{Q}}^2 = \frac{p^2}{p_0^2}$ and thus $0 < \dot{\mathbf{Q}}^2 < 1$.

Moreover, by solving Equation (3) for $\mathbf{Q}^{(t)}$, the solution

$$\mathbf{Q}^{(t)} = \mathbf{Q} + \begin{bmatrix} \frac{\mathbf{p}}{p_0} & 0 \\ 0 & -\frac{\mathbf{p}}{p_0} \end{bmatrix} t \quad (29)$$

is obtained, which is free from singular behaviours.

**Remark 2.** *With regard to Remark 1, we see that (28) leads to the relativistic kinetic energy operator $E_{Kin} = \frac{\mu}{\sqrt{1-\dot{\mathbf{Q}}^2}} \equiv p_0$, which is different from $P_0 = \begin{bmatrix} p_0 & 0 \\ 0 & -p_0 \end{bmatrix}$. Thus, negative values of $\sigma(P_0)$ do not entail a negative kinetic energy.*

## 6. Characterization of Dirac's Theory by Peculiar Transformation Properties

Theorem 2 implies that, fixing $\mu > 0$ and any positive spin $s$, including $s = 1/2$ as in Dirac's theory, there are six classes of possible inequivalent theories of elementary free particles satisfying (23), one for each inequivalent triplet identified by (16). A particular theory in each class is characterized by a particular choice of the pair $\eta, \gamma$ satisfying the conditions established by Theorem 2.

In correspondence with a particular theory there is a different relation for the transformation properties of the position with respect to boosts, i.e., a different relation for the commutators $[K_j, Q_k]$. The validity of each theory depends on the validity of such a relation.

**Example 1.** *Let us consider the possible theory based on a triplet of $\mathcal{I}^{-+}(\mu, s)$ with $^{\triangleleft}T_1$ and $_{\mathscr{S}}S_1$ in (16) as time-reversal and space-inversion operators. According to theorem 5.1, if $\gamma = 0$ is chosen, then*

$$\mathbf{Q} = \begin{bmatrix} \mathbf{F} & 0 \\ 0 & \mathbf{F} \end{bmatrix}. \quad (30)$$

*This means that in this particular theory,*

$$[K_j, Q_k] = \begin{bmatrix} [K_j, F_k] & 0 \\ 0 & -[K_j, F_k] \end{bmatrix}. \quad (31)$$

*The explicit relation for $[K_j, F_k]$ can be computed by making use of (14) and (15), with the following result, where $(j, k, l)$ is a cyclic permutation.*

$[K_j, F_k] = -\frac{p_j p_k}{p_0^2} \frac{1}{p_0} + \frac{p_k}{p_0} \frac{\partial}{\partial p_j} + i \frac{S_l}{\mu + p_j} - i \frac{S_l p_k - S_k p_l}{(\mu + p_0)^2 p_0}$, *if $j \neq k$, if $j, k$;*

$[K_j, F_j] = \frac{1}{2p_0} - \frac{p_j^2}{p_0^2} \frac{1}{p_0} + \frac{p_j}{p_0} \frac{\partial}{\partial p_j} - i \frac{S_k - S_l) p_j p_k}{(\mu + p_0)^2 p_0}$.

*This theory is valid if and only if these relations are valid.*

Moreover, Dirac's theory is characterized by a peculiar transformation property of the position with respect to boosts. Since the zitterbewegung is a peculiar feature of Dirac's theory, expressed by (22), Dirac's theory cannot belong to the alternative classes identified in Section 4 for which (29) holds. Therefore, we expect that the transformation properties of $\mathbf{Q}^D$ with respect to boosts are different from those of the alternative theories. Let us show that this is in fact the case.

Jordan and Mukunda [23] indeed proved that in a triplet of $\mathcal{I}^{-+}(\mu, 1/2)$, the position operator of Dirac's theory is

$$\mathbf{Q}^D = \begin{bmatrix} \mathbf{F} + \frac{\mathbf{p} \wedge \mathbf{S}}{p_0(\mu+p_0)} & -i\frac{(\mathbf{p}\cdot\mathbf{S})\mathbf{p}}{p_0^2(\mu+p_0)} + i\frac{\mathbf{S}}{p_0} \\ i\frac{(\mathbf{p}\cdot\mathbf{S})\mathbf{p}}{p_0^2(\mu+p_0)} - i\frac{\mathbf{S}}{p_0} & \mathbf{F} + \frac{\mathbf{p}\wedge\mathbf{S}}{p_0(\mu+p_0)} \end{bmatrix}. \tag{32}$$

If the commutator $[K_j, Q_k^D]$ is computed, then the following result is found.

$$[K_j, Q_k^D] = -\frac{i}{2}(Q_j^D \dot{Q}_k^D + \dot{Q}_k^D Q_j^D). \tag{33}$$

It is very interesting that (33) completely characterizes Dirac's theory; indeed, in [19], it was proven that a commutative three-operator $\mathbf{Q}$ satisfying (17) and (18) in a theory based on a triplet of $\mathcal{I}^{-+}(\mu, s)$, with $s > 0$, also satisfied (33) if and only if it had the form (32). Thus, Dirac's theory is the unique theory that satisfies $\mathcal{I}P$ and the covariance properties with respect to translations and rotations, 𝕥 and 𝕤, whose transformation properties of the position with respect to boosts are expressed by (33).

### 7. The Problem of the Experimental Comparison of the Theories

Our purely theoretical investigation is not able to establish which of the possible theories for Dirac particles here identified is the "right" theory. In fact, we have no further theoretical criterion that helps the choice; in particular, a general explicit condition establishing the transformation properties of the position with respect to boosts is not available. This indeterminacy could be solved by experiments; this option entails identifying experiments such that the results predicted for each of them by the different theories considered are different. Then, the performance of these experiments would allow to single out which theory, if any, predicts the actually obtained outcome, i.e., the empirically valid theory.

Very often, the derivation of these predictions cannot be obtained when working within a theory of free particles. As a historical example, the first success of Dirac's theory can be pointed out, i.e., the experimental discovery of the positron, the antiparticle of the electron, whose existence is predicted by Dirac's theory as a Dirac particle with state vector $\begin{bmatrix} 0 \\ \psi \end{bmatrix}$ in our representation based on $\mathcal{I}^{-+}(\mu, 1/2)$. Now, a positron differs from an electron by the positive sign of its electrical charge, so that it can be distinguished from the electron because their paths have opposite curvatures in a cloud chamber in the presence of a uniform magnetic field. In order to theoretically, as well as quantitatively, predict these behaviours, the theory of a particle interacting with an electromagnetic field is needed. Dirac attained this theory by replacing the operator $i\frac{\partial}{\partial t}$ with $i\frac{\partial}{\partial t} + e\phi(\mathbf{Q}^D)$ and $\mathbf{P}^D$ with $\mathbf{P}^D - e\mathbf{A}(\mathbf{Q}^D)$ in his Equation (21), where $\phi$ and $\mathbf{A}$ are the electromagnetic potentials; for a uniform magnetic field $\phi = 0$ and $\mathbf{A} = (B_0 x_3, 0, 0)$. The theory so obtained predicted curvatures that turned out to be confirmed by experimental observation.

The analogous prediction within one of the alternative theories derived by the present approach cannot be carried out, because they are theories of *free* particles; in fact, the theory of a particle interacting with an electromagnetic field should be derived by complying with the present methodological commitment that requires a strictly deductive development from the basic principles. This development was successfully achieved for the nonrelativis-

tic case [24,25], but it is yet to be accomplished for a relativistic theory. The Dirac method, i.e., to replace $i\frac{\partial}{\partial t}$ with $i\frac{\partial}{\partial t} + e\phi(\mathbf{Q}^D)$ and $\mathbf{P}$ with $\mathbf{P} - e\mathbf{A}(\mathbf{Q}^D)$ in one of the alternative free-particle theories, for instance in the theory of example 6.1, leads to the wave equations $i\frac{\partial \psi_t}{\partial t} = \sqrt{\mu^2 + (p_1 - eB_0 x_3)^2 + p_2^2 + p_3^2}\,\psi_t$ and $i\frac{\partial \psi_t}{\partial t} = -\sqrt{\mu^2 + (p_1 - eB_0 x_3)^2 + p_2^2 + p_3^2}\,\psi_t$ for electron states $\begin{bmatrix} \psi_t \\ 0 \end{bmatrix}$ and positron states $\begin{bmatrix} 0 \\ \psi \end{bmatrix}$, respectively. These are standard equations for a charge in a uniform magnetic field, whose nondifficult solutions predict the observed values of the curvatures. We have to reiterate, however, that the equations used are not derived according to the methodological commitment adopted in the present work.

For an experiment that involves an interacting particle, since a theory of an interacting Dirac particle alternative to Dirac's theory is missing, there is no alternative prediction to be experimentally tested. Therefore, the empirical domain allowing for a comparison is restricted to experiments with a free particle. For a free particle, the difference between the predictions of Dirac's theory and those of an alternative one, say the theory of Example 1, is the existence of the zitterbewegung denied by the alternative theory. Now, it has been theoretically evaluated [5,9,26] that the oscillatory behaviour of the zitterbewegung has an amplitude of the order of the Compton wavelength, while its characteristic frequencies are numerically about $2m_0 c/\hbar$ ($m_0$ is the rest mass of the electron). These extreme values make the experimental observation of the zitterbewegung directly on a free particle extremely difficult. However, different physical consequences, observable at least in principle, can be derived by the two different theories. In Section 7.1, we set up the formalism where the different predictions can be derived. In Section 7.2, the conditions are identified, which lead to different predictions observable at least in principle.

*7.1. The Formalism for the Different Predictions*

According to Dirac's theory, as represented in Section 4.1, since $\dot{\mathbf{Q}}^D = \boldsymbol{\alpha}$, the time evolution of the position operator is given by

$$\dot{\mathbf{Q}}^D(t) = \mathbf{Q}^D + \boldsymbol{\alpha}t + \mathbf{o}(t) = \mathbf{x} + +\boldsymbol{\alpha}t + \mathbf{o}(t), \tag{34}$$

where $\mathbf{o}(t)$ is an Hermitian three-operator infinitesimal of order greater than one with respect to time $t$. The time evolution of the position operator according to the alternative theory of Example 1 is given by

$$\dot{\mathbf{Q}}(t) = \mathbf{Q} + \frac{\mathbf{P}}{P_0}t; \tag{35}$$

in order to make the comparison with (34) more direct, we reformulate the alternative theory in the following equivalent form, by means of a unitary transformation operated by the unitary operator $Z : L_2(\mathbb{R}^3, \mathbb{C}^4, dv) \rightarrow L_2(\mathbb{R}^3, \mathbb{C}^4)$, with $Z = Z_1 Z_2$, where $Z_2 = \frac{1}{p_0}$, while $Z_1$ is the inverse Fourier–Plancherel operator. In so doing, the Hilbert space of the reformulated alternative theory coincides with the Hilbert space of Dirac's theory. The operators $\mathbf{Q}$ and $P_j$ become $\mathbf{Q} = \mathbf{x} = \mathbf{Q}^D$ and $P_j = -i\frac{\partial}{\partial x_j} = P_j^D$. The operator $P_0$ becomes

$$P_0 = \begin{bmatrix} \sqrt{\mu^2 - \boldsymbol{\nabla}^2} & 0 \\ 0 & -\sqrt{\mu^2 - \boldsymbol{\nabla}^2} \end{bmatrix}. \text{ Thus, (35) becomes}$$

$$\mathbf{Q}(t) = \mathbf{x} + \frac{\mathbf{P}}{P_0}t. \tag{36}$$

Single measurements of the position at time $t$ cannot establish whether (34) or (36) is valid, of course. We show in the next subsection that the validity of (34) or (36) can be settled through the experimental evaluation of the different expectation values corresponding to (34) and (36).

*7.2. The Different Ideally Observable Predictions*

Let $\psi^D(\mathbf{x}) = \frac{1}{2} \frac{e^{-\frac{x^2}{4\varsigma_0^2}}}{(2\pi\varsigma_0^2)^{\frac{3}{4}}} \begin{bmatrix} 1 \\ 1 \\ 1 \\ 1 \end{bmatrix} = \frac{1}{2}\varphi(\mathbf{x}) \begin{bmatrix} 1 \\ 1 \\ 1 \\ 1 \end{bmatrix}$ be the state vector of a free Dirac particle

according to Dirac's theory, where $\varphi(\mathbf{x}) = \frac{e^{-\frac{x^2}{4\varsigma_0^2}}}{(2\pi\varsigma_0^2)^{\frac{3}{4}}}$ is the normalized gaussian wave function of expectation value zero and standard deviation $\varsigma_0$.

The position probability density is $\rho_q^D(\mathbf{x}) = |\varphi(\mathbf{x})|^2 = Ne^{-\frac{x^2}{2\varsigma_0^2}}$, while the $\mathbf{P}^D$ probability density is $\rho_p^D(\mathbf{p}) = |\hat{\varphi}(\mathbf{p})|^2 = Me^{-\frac{p^2}{2\varsigma_1^2}}$, where $\hat{\varphi}$ is the Fourier–Plancherel transform of $\varphi$, so it is a Gaussian of expectation value zero and standard deviation $\varsigma_1 = \frac{1}{2\varsigma_0}$, $N$ and $M$ being normalization constants.

Let us compute the expectation value of $\mathbf{Q}^D(t)$ determined by the state vector $\psi^D$. By making use of (34), we find $\langle Q_j^D(t)\rangle = \langle x_j \rangle + \langle \alpha_j \rangle t + o_j(t)$, where $o_j(t)$ is a function infinitesimal of order grater than 1 with respect to $t$. Since $\rho_q^D(\mathbf{x})$ is even, we have $\langle x_j \rangle = 0$; on the other hand $\langle \alpha_j \rangle = \langle \psi^D \mid \alpha_j \psi^D \rangle = \frac{1}{4}[1 \quad 1 \quad 1 \quad 1]\left( \alpha_j \begin{bmatrix} 1 \\ 1 \\ 1 \\ 1 \end{bmatrix}\right)$, so since $\alpha = \begin{bmatrix} 0 & \sigma_j \\ \sigma_j & 0 \end{bmatrix}$,

we have $\langle \alpha_1 \rangle = 1$, $\langle \alpha_2 \rangle = \langle \alpha_3 \rangle = 0$. Therefore,

$$\langle Q_1^D(t)\rangle = t + o_1(t), \quad \langle Q_2^D(t)\rangle = o_2(t), \quad \langle Q_3^D(t)\rangle = o_3(t). \tag{37}$$

In order to compute the expectation value of $\mathbf{Q}(t)$ in (36), we need the state vector $\psi$ that, in the alternative theory, represents the quantum state represented by $\psi^D$ in Dirac's theory. This $\psi$ is unknown. However, the position operator and the self-adjoint generators of translations coincide in the two theories:

$$\mathbf{Q}^D = \mathbf{x} = \mathbf{Q}, \quad \mathbf{P}^D = -i\boldsymbol{\nabla} = \mathbf{P};$$

moreover, the operators $\mathbf{x}$ and $-i\boldsymbol{\nabla}$ have the same physical meaning in the two theories; therefore, the probability densities determined by $\psi^D$ and the unknown $\psi$ must coincide as well in the two theories:

$$\rho_q^D(\mathbf{x}) = \rho_q(\mathbf{x}) = Ne^{-\frac{x^2}{2\varsigma_0^2}}, \quad \rho_p^D(\mathbf{p}) = \rho_p(\mathbf{p} = Me^{-\frac{p^2}{2\varsigma_1^2}}.$$

The knowledge of $\rho_q$ and $\rho_p$ allows us to compute the expectation value of $\mathbf{Q}(t)$ in (36); since both densities are even functions and both terms in (36) are odd, we obtain

$$\langle \mathbf{Q}(t)\rangle = \mathbf{0}. \tag{38}$$

We see that that the predictions (37) and (38) are different. In particular, according to (37), a time $t_0 > 0$ must exist such that $\langle \mathbf{Q}_1^D(t)\rangle > 0$ for all $t$ such that $0 < t < t_0$, whereas $\langle \mathbf{Q}_1^D(t)\rangle = 0$ for all $t$. Hence, if Dirac's theory is valid, by measuring the mean values of $Q_1^D$ in correspondence to a sequence of times $t_n = \frac{1}{2^n}$, after a given $n_0$, all mean values must be coherent with positive expectation values. The alternative theory, on the contrary, predicts that all mean values must be coherent with a null expectation value. Thus, we have different predictions that, in principle, can be experimentally tested.

Of course, the experimental test here described has a quite ideal character. In particular, serious difficulties occur, for instance, in order

**(a)** To implement the experimental conditions such that the state vector of the particle is $\psi^D$ according to Dirac's theory;

**(b)** To perform measurements of the particle coordinate $x_1$ at extremely small, fixed times $t_n$ to evaluate the expectation values, without introducing a noninstantaneous interaction.

Yet, the present argument shows that there are physical effects that distinguish Dirac's theory from alternative theories.

## 8. Discussion

We have addressed the development of the quantum theory for a Dirac particle following a strictly deductive method, which derives the theory from the basic principles specifying a free particle, i.e., from a theoretical invariance and position covariance with respect to the transformation of the Poincaré group $\mathcal{P}$, by making use of effective group-theoretical methods. In so doing, we have found that Dirac's theory is not the unique theory of a Dirac particle consistent with the basic principles; in fact, for every irreducible transformer triplet $(U, \mathcal{S}, {}^\triangleleft\mathrm{T})$ in $\mathcal{I}^{-+}(\mu, 1/2)$, there is a class of consistent possible theories; each specific theory in one of these classes should be identified by the specific position operator $\mathbf{Q}$ of that theory. Hence, to identify the position operator, $\mathbf{Q}$ should be asked to satisfy the covariance properties according to ($Q$.2). By requiring the covariance properties with respect to translations, rotations, time reversal and space inversion, constraints on $\mathbf{Q}$ can be established, but without completely determining it. The transformation properties with respect to boosts are not available, therefore they cannot be used to better identify $\mathbf{Q}$; as a consequence, for every triplet in $\mathcal{I}^{-+}(\mu, 1/2)$, there is a subclass of possible theories; a theory in one such subclass, identified by a given $\mathbf{Q}$, is valid only if the transformation property with respect to boosts determined by computing $[K_j, Q_k]$ is valid in nature. Therefore, in example 6.1, if the relations found for $[K_j, Q_k]$ are valid, then the theory based on the first triplet in (16) with $\mathbf{Q} = \begin{bmatrix} \mathbf{F} & 0 \\ 0 & \mathbf{F} \end{bmatrix}$ is valid. For Dirac's theory, if the relation (33) is valid, then Dirac's theory is valid with $\mathbf{Q}$ given by (32).

Thus, due to the lack of a general transformation property of the position with respect to boosts, consistency with the basic principles is not sufficient to identify a unique theory of a Dirac particle. The possible theories identified by (23) have the same "degree of validity" as Dirac's theory. The experimental comparison of the theories should help solve this indeterminacy. For the time being, this comparison can be realized only with respect to predictions of free-particle theories. At this level, we have derived different predictions by different theories. The difference, which only concerns the zitterbewegung, is observable in ideal experiments.

**Funding:** This research received no external funding.

**Data Availability Statement:** Not applicable.

**Conflicts of Interest:** The author declares no conflict of interest.

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
