# Peer review of "Consistent Theories of Free Dirac Particle without Singular Predictions"

_2571-712X, doi:10.3390/particles6010001_

Round 1

Reviewer 1 Report

The manuscript by Giuseppe Nisticò is about   alternative to the Dirac theories of the relativistic electron.  The author based on Group theory motivates his work by some fundamental issues such as the  lack of satisfying position operator  that in the standard theory leads to   the well known zitterbewegung effect.  While I am not able  to judge all the technical aspects of that work  the claims and proposals are interesting and could certainly motivate further research in that direction.   I would therefore suggest publication unless a different report written by an expert on the filed would find an error in the analysis of the author.

Author Response

Dear Reviewer,

your report suggested publication unless another expert woulf find an error 

in the analysis of the author. Now, Reviewer 3 also suggested publication, while reviewer did not find errors in the analysis, but he rised a criticism because  an account on the experimental comparison of the alternative theories with Dirac theory  is missing in the article. Agreeing with this criticism I accordingly have the article. My answer is in the attached file.

Best regards

Giuseppe Nisticò 

Reviewer 2 Report

This is a study in mathematical physics, which tries to elucidate the possible structure of spin-1/2 free particle theory. It concludes that the well known Dirac theory is not the only possible spin-1/2 theory consistent with  certain basic requirements. In passing the author criticizes the well known canonical quantization procedures as "not deductive".

While this is true I suggest that the relevant judge if a theory is relavant, consistent useful or valid is EXPERIMENT, as physics is an experimental science. Without experiment to judge the predicted results of a theory, we a talking about mathematics only without any semantics. Without any doubt Dirac theory has proven its relevance and consistency in countless applications. By the way, Zitterbewegung which is indeed predicted by Dirac theory is not ruled out by experiment, it is just not seen (yet). Some experiments have been made suggesting that it has been already seen.

So, Zitterbewegung may indeed be a valid and experimentally verifiable consequence of the theory, how singular it may appear.

Nevertheless, the author suggests a theory which does not predict Zitterbewegung, therefore it is definitely different from standard Dirac theory.

However, the author does not seem to address the question, if the proposed theory is consistent with all other tested applications of Dirac theory. Without a detailed address of this question, the paper remains purely formal, and of little value to PHYSICS.

In its present form I cannot recommend the paper for publication. I would suggest the auther tries to elucidate the question in what sense the new theory complies with all successes of Dirac theory without introducing new problems. Eventually with suggestions on how to test the new theory experimentally.

Author Response

Dear Reviever,

I acknowledged that the article lacks an account about the experimental validation of the theories at issue, and I have acoordingly modified the article. My detailed answer is in the attached file.

Best Regards

Giuseppe Nisticò

Reviewer 3 Report

This paper is a very interesting discussion of alternative approaches to Dirac theory. It is well written and of high interest. I recommend that it be published.

Author Response

Dear Reviewer,

Your report recommended publication apart from a minor spell check that has been performed, now. 

I have modified the paper accordin to the reports of the other reviewers.

Best regards

Giuseppe Nisticò

Round 2

Reviewer 2 Report

I am very pleased that the author decided to add section 7 to the manuscript motivated by the questions posed in my previous report.  It adds an important element to the discussion, and emphasizes that the expectation values of the quantities (Q^D)^2  and  (Q)^2 differ from each other (where (Q^D) is the position operator in Dirac theory and Q the position operator in the alternative theory discussed by the author).

However, to be of relevance to an "experimental validation" as suggested by the section heading, it is not enough to show that one can calculate expectation values which differ in different theories. It would be imperative to suggest at least in principle how to measure such expectation values, and here in particular without introducing any interactions.

So, for me the fundamental question how to distinguish between the different theories discussed by the author is not (yet) answered fully satisfactorily.

As an aside: I do not understand what the author wants to say after the equation in section 7 which shows the difference between the two expectation values citing Barut and Malin ("one gets essentially the same result if one uses wave packets instead of p=0 vector states.")

I cannot really recommend yet this paper for publication before the author addresses these issues.

Author Response

Dear Reviewer,

I have revised the article taking into account your comments.

The attached fie contains the detailed answer.

Sincerely

Giuseppe Antonio Nisticò

Round 3

Reviewer 2 Report

I appreciate that the author carefully considers the referee's comments, which makes for an interesting discussion. It now appears that the paper should be discussed by a wider audience. Therefore, I recommend the paper for publication despite my feeling that there are quite a number of questions and issues remaining.

Nevertheless I recommend once again a careful proofreading. In particular in the new sections I find a number of spelling mistakes (e.g.  line 421 "interancting" 436 "sigle" or  eq. (34 )  double plus sign).